# A comprehensive scoping review of intergenerational dance programmes for cohorts with a generational gap

Siobhán O'Reilly[1,2]*, Orfhlaith Ní Bhriain[2,3,4], Sarah Dillon[1,2], Amanda M. Clifford[1,2,3]

1 School of Allied Health, University of Limerick, Castletroy, Limerick, Ireland, 2 Health Research Institute, University of Limerick, Castletroy, Limerick, Ireland, 3 Ageing Research Centre, University of Limerick, Castletroy, Limerick, Ireland, 4 Irish World Academy of Music and Dance, University of Limerick, Castletroy, Limerick, Ireland

* oreilly.siobhan@ul.ie

**Data Availability Statement:** All relevant data are within the paper and its Supporting Information files.

## Abstract

### Introduction

Loneliness and physical inactivity are issues that affect both young people and older adults. This can have negative health outcomes and well as high costs on health services. Physical activity can positively influence both physical and psychosocial health outcomes, however enjoyment is necessary for adherence. Combining exercise with arts-based activities can improve enjoyment for older adults and young people. Dance has been found to be a safe and enjoyable form of physical activity that can be equally or more effective than conventional exercise options. Intergenerational interventions can improve relationships between generations. The aim of this scoping review was to collate and map the available evidence for intergenerational dance.

### Methods

This scoping review followed the guidance outlined by the Joanna Briggs Institute. A systematic search of nine multidisciplinary databases and four repositories was conducted. Inclusion criteria were intergenerational dance or movement to music programmes. Exclusion criteria included dance movement therapy or groups with less than one generational gap. Data were extracted and summarised using narrative synthesis and research papers were appraised using the Mixed Methods Appraisal Tool.

### Results

The search identified eleven research studies, seven expert opinion/practice expertise and 13 sources from the grey literature. Dance classes were typically 11–12 weeks long. Genres varied with some programmes including co-creation through choreography. Experiences and social outcomes were the most assessed outcomes, with a lack of studies examining physical outcomes. Participants reported enjoying the programmes stating they felt proud for taking part and looked forward to sessions. The term *intergenerational* was not defined in any paper.

**Funding:** Award: Irish Research Council Government of Ireland Postgraduate Scholarship Grant number: GOIPG/2023/4704 Received by: SO'R Name of funder: Irish Research Council URL: https://research.ie/ Award: Evidence Synthesis Fellowship Grant number: CBES-2018-001 Received by: SD Name of funder: Irish Health Research Board and HSC Public Health Agency in association with Evidence Synthesis Ireland/ Cochrane Ireland URL: https://www.hrb.ie/ The sponsors did not play any role in study design, data collection and analysis, decision to publish, or preparation of the manuscript.

**Competing interests:** The authors have declared that no competing interests exist.

## Conclusions

Intergenerational dance is an emerging area of research. Many programmes run in communities but are not researched, therefore several gaps remain. More large-scale trials are needed around intergenerational dance. Definitions and descriptions of dance and intergenerational activity should be considered in future studies to ensure consistency.

## Introduction

Physical inactivity, social isolation and loneliness are key health issues across the lifespan and particularly in older adults and adolescents [1,2]. Physical inactivity and loneliness have been identified as leading factors for mortality in adult populations [3–5]. Physical inactivity is linked to over 13 million disability-adjusted life-years globally [6]. Over one-fifth of older adults do not meet the World Health Organisation recommended physical activity (PA) guidelines [7]. For older adults, sedentary behaviours have been found to be associated with social isolation and depressive symptoms [8,9] with secondary effects on physical health, specifically cardiovascular health [5]. The majority of teenagers self-report insufficient levels of PA [10], and higher than recommended screen time [11]. Sedentary behaviour and screen time are associated with lower levels of social connectedness in adolescents [12], which can worsen mental health and increase the risk of obesity [10]. It is estimated that the costs associated with physical inactivity is upwards of millions of dollars, from both direct and indirect costs [13]. Similarly, loneliness has been found to have indirect costs linked to increased general practitioner and mental health services [14]. Hence social opportunities that encourage regular and consistent PA are needed to offset this burden.

Several national, European, and international policies and frameworks have been put in place to promote PA and tackle loneliness [15–18]. Internationally, the United Nations Decade of Healthy Ageing (2021–2030) framework aims to promote healthy ageing [15] through combating ageism, and creating age-friendly environments to enable older adults to continue participating in activities they enjoy. Multiple Irish frameworks aim to increase physical activity levels through strategies such as education, public awareness and encouraging young people and their families to start good habits young [17,18]. For children and adolescents, a guiding framework has been outlined to increase PA in schools, co-curricular activities, and in the community [16]. Despite these policies and frameworks, physical inactivity and loneliness continue to be key concerns in the global population.

Group-based exercise interventions often have social elements, creating a synergy between socialisation and PA [19]. However, barriers such as the increasing competitive nature in children's sports, along with parental pressure, lack of enjoyment and costs lead to more and more young people dropping out of sports [20]. For older adults, barriers include concerns of falling and discomfort during exercise [21]. In contrast, viewing exercise as a positive and enjoyable experience is a motivator for older adults and youths alike when joining a sport or being physically active [21,22]. There is a need for community-based opportunities for older adults and youths to engage in enjoyable forms of physical activity that promotes engagement, fosters a sense community and increase social connectedness [23–25]. Research has shown that increased enjoyment can be achieved by combining exercise with arts-based activities for children and adolescents and older adults [26,27]. Additionally, engaging in arts-based activities can improve social supports for adolescents [28] and older adults [29,30] alike. There is a need for fun inclusive forms of exercise that can engage both older adults and young people.

Dance is an artistic and social form of PA that has been found to benefit older adults and young people and may be an enjoyable alternative to conventional exercise interventions. A review comparing dance over all age groups and cohorts (children, adults, older adults, people with illnesses/disabilities) found that dance is equally effective, and in some cases more effective, than conventional exercise programmes [31]. Dance interventions have been found to be safe and acceptable [32,33], with generally high adherence rates and can have positive effects on mobility and endurance [32], health-related quality of life [33], and cognition [34]. Similarly, in children and adolescents, dance has been found to be an acceptable, enjoyable and equally effective or superior alternative to conventional PA interventions, while providing physiological benefits for children [35,36]. Dance interventions are typically held in safe community spaces and are free of charge, providing opportunities for intergenerational bonding [35].

While older adults and young people differ in many respects, they share certain social issues, such as the peaks of loneliness in young and old age [37] and an increase in sedentary behaviour [38,39]. Intergenerational dance may be an enjoyable alternative to increase physical activity and socialisation for those who are not interested in conventional sports or exercise interventions [40]. However, intergenerational dance programmes are largely unexplored despite their potential to target the shared issues within both groups. To date, no published review has collated the evidence on intergenerational dance. The aim of this scoping review is to identify and map the literature on dance programmes for intergenerational cohorts. As improvements have been shown through dance for older adults and young people separately, it is important to explore the benefits such programmes can have on combined groups. It is hoped that the findings from this review may promote future research in intergenerational dance that will address the identified gaps.

The research question for this review is *what is the evidence for intergenerational dance programmes with an intergenerational gap between participants*? Therefore, the objectives are: 1) Map the current available empirical evidence on intergenerational dance programmes; 2) Identify the evidence gaps that require further research; 3) Identify terms used to define "intergenerational" for older adult/child or adolescent dyads in the dance programmes.

## Methods

Scoping reviews are used to map emerging evidence in an area to give an overview of the extent and content of the literature [41]. Following an initial search, no pre-existing reviews were found in this area. A scoping review was considered to be appropriate to identify the type of available evidence for intergenerational dance the term "intergenerational" was explored and gaps were identified. This review was reported in accordance with the Preferred Reporting Items for Systematic reviews and Meta-Analyses extension for Scoping Reviews (PRISMA-ScR) checklist [42] which can be found in S1 File. The methodology of this paper follows the Joanna Briggs Institute (JBI) guidance for conducting scoping reviews [31]. This framework was adopted to ensure the review would be replicable and rigorous. This was based on the guidance originally published by Arksey and O'Malley [43] and later updated by Levac *et al*. [44]. A protocol was written for this review and registered on Open Science Framework [45]. Ethical approval was not required for this review.

The inclusion criteria were outlined using the PCC framework as recommended by the JBI guidelines [46]. The population was: Intergenerational, with at least one generational gap. For the concept, all dance or movement to music programmes for intergenerational groups were considered eligible for inclusion. The context was all settings.

Intergenerational activity had to be part of the primary intervention of the studies. Coincidental age-mixing was not considered for inclusion. The exclusion criteria were dyads with

less than one generational gap. Any programme involving Dance Movement Therapy was excluded as this is a form of psychotherapy and a decision was made to focus on non-therapeutic dance programmes. Regarding grey literature (websites) one-off events were excluded.

### Types of sources

All types of literature were eligible for this scoping review. These included, but were not limited to, all forms of empirical experimental or quasi-experimental studies, qualitative studies, mixed methods studies, and non-empirical reviews such as systematic reviews or umbrella reviews. Grey literature was also eligible for inclusion. No limits were set for year of publication or language.

### Search strategy

An example search strategy can be found in S2 File. The search was conducted in October 2023. The systematic approach to searching outlined by Peters, Godfrey [31] was used. This involved three steps: a preliminary search to identify keywords, a second search using these keywords, and finally a search of the reference lists of the selected papers. A full search was developed following preliminary searches in CINAHL and PubMed Central, discussions with interdisciplinary members of the research team and input from an experienced research librarian with expertise in the area. To capture the interdisciplinary nature of this review, a wide range of relevant databases were searched including CINAHL Complete, AMED, PubMed Central, PsycINFO, Cochrane-CENTRAL, Scopus, Web of Science, Performing Arts periodicals Database, and Applied Social Sciences Index & Abstracts (ASSIA). The following repositories were checked for grey literature: Arts Council of Ireland, National Library of Ireland, Digital Repository of Ireland, and the Library of Trinity College Dublin. Additionally, Google Advanced and Scopus were checked for grey literature. Finally, Google Scholar, limited to the last year, was checked to identify recent papers not yet indexed in a database. A Microsoft Excel sheet was created to track the results from the searches, with each search term tracked individually and then in combination.

### Study/Source of evidence extraction

The results of the searches were uploaded to the Rayyan software [47], where duplicates were removed. Two reviewers (S. O'R. and S.D.) independently screened the titles and abstracts of the papers and conflicts were resolved via discussion. When a resolution was not made, a third independent reviewer (A.C.) was involved to determine eligibility of the papers. The full texts of the selected papers were sourced and assessed in full for eligibility. If the full text was not available, the authors were contacted via email for more information. The same method for resolving conflicts through discussion was used for these papers. The papers to be included in the review were exported to the reference manager EndNote 21 [48]. The PRISMA flow diagram was used to display the study selection journey [49]. This can be found in Fig 1.

Critical appraisal [50] was conducted to provide information on the quality of relevant empirical studies using The Mixed Methods Appraisal Tool (MMAT) [51] and to highlight relevant studies to inform future studies in the area [52]. The MMAT was chosen as it allowed for several types of evidence to be appraisal using one tool (mixed methods, qualitative, quantitative randomised control trial, quantitative non-randomised, and quantitative descriptive). No study was excluded based on quality. This was completed by the lead author (S. O'R.) and a random sample of 50% was reviewed by a second researcher (S.D.) to ensure agreement.

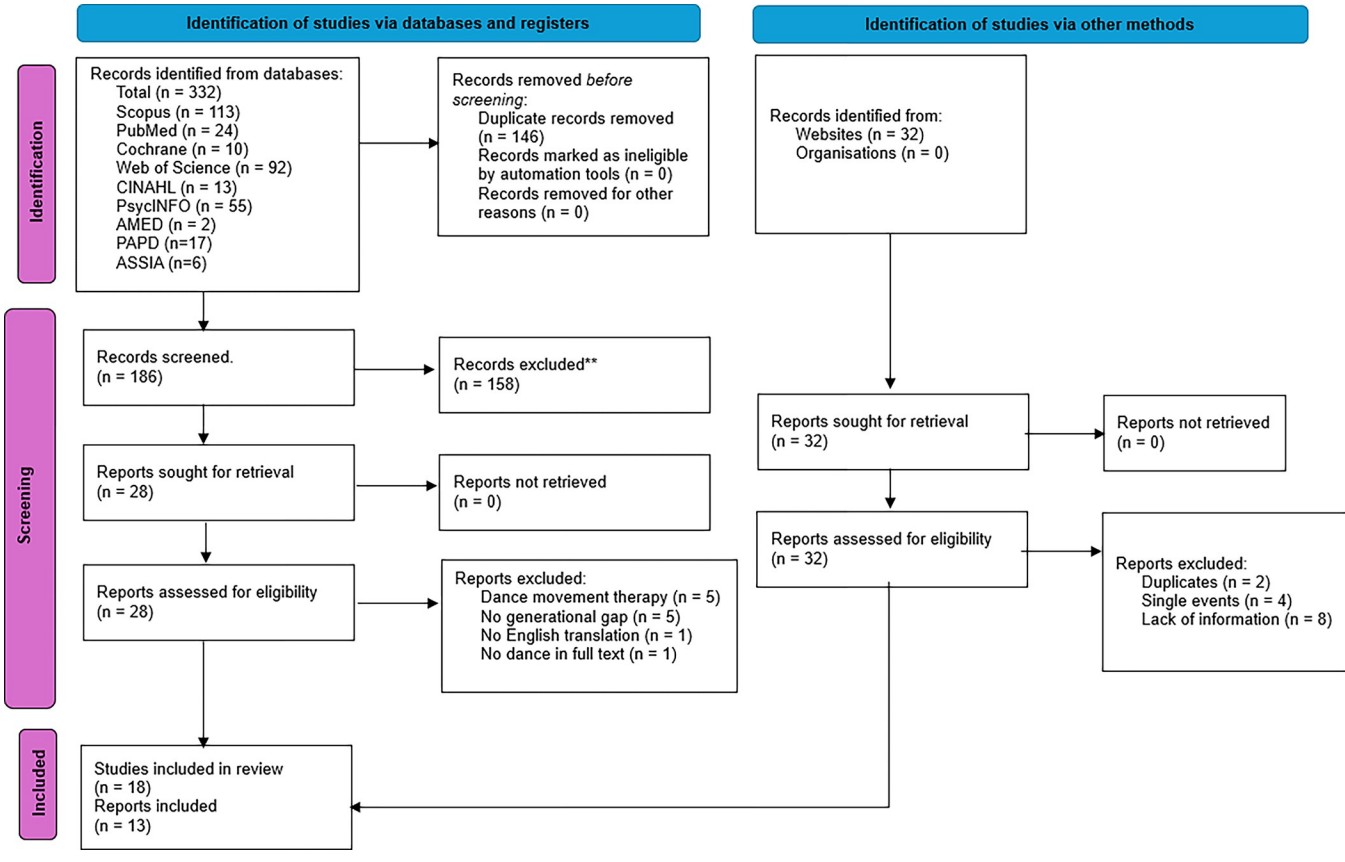

**Fig 1. PRISMA flow diagram.**

## Data analysis and presentation

Data were extracted by the lead author. The full data extraction table can be found in S1 Table. S.D. cross-checked a proportion of 20% of these papers. The data were extracted into three tables, divided into interventional studies, expert opinion/practice expertise research papers, and community classes. The common data extracted included year, country, setting, publication type, older and younger cohort descriptions and *intergenerational* definition. More detail was extracted from the research studies such as outcome measures, control, frequency, intensity, time and type (FITT) principles and results. The results were summarised based on the type of literature (research studies, expert opinion/practice expertise, or grey literature). Qualitative data from the mixed methods and qualitative papers were synthesised using a descriptive narrative approach. The framework outlined by Popay, Roberts [53] was followed. This involved synthesising the data by identifying similarities and differences between the papers. Patterns were recognised and these were categorised under different headings. While this method is typically used in systematic reviews, it has been used in scoping reviews previously [54–56]. Finally, gaps in the literature were identified and narratively synthesised.

## Results

The included literature comprised eighteen research evidence papers (eleven interventional studies [57–67] and seven expert opinion/practice expertise papers) and thirteen sources from the grey literature. Table 1 outlines the different source types included. Four of the research

**Table 1. Evidence type.**

| Evidence Type | Description | | Papers |
|---|---|---|---|
| Research studies (n = 11) | Quantitative (n = 2) | Quantitative observational study (n = 1) | Morita and Kobayashi, 2013 [59] |
| | | Pretest-posttest trial (n = 1) | Rossberg-Gempton and Poole, 2000 [65] |
| | Qualitative (n = 5) | Case study (n = 1) Methods: Interviews and field logs | Sherman, 1997 [58] |
| | | Qualitative inquiry (n = 1) Methods: Semi-structured focus groups | Wu *et al.* 2023 [61] |
| | | Qualitative study (n = 1) Methods: Focus groups with constructivist approach | Farrer *et al.* 2022 [62] |
| | | Descriptive exploratory study (n = 1) Methods: Semi-structured focus groups | Jenkins *et al.* 2021 [63] |
| | | Video recording with observation with focus groups (n = 1) | Rossberg-Gempton *et al.* 1999 [66] |
| | Mixed Methods (n = 4) | Pretest-posttest trials with qualitative interviews (n = 3) | Douse *et al.* 2020 [57] Brandão *et al.* 2021 [60] Young *et al.* 2014 [67] |
| | | Nested mixed methods study (n = 1) Methods: Survey with quantitative and qualitative elements | D'Cunha *et al.* 2023 [64] |
| Expert opinion/ Practice expertise (n = 7) | Expert opinion on intergenerational choreography (n = 2), Dance experts/choreographer practice expertise (n = 2), Book chapters by researchers and dancers (n = 3) | | Prichard, 2022 [68]; Corbin, 1998 [69]; Stock, 2018 [70]; Amans, 2007; Hopfinger, 2018 [71]; Warner, 2012 [72]; Dreyfuss, 2016 [73]. |
| Grey literature (n = 13) | Websites detailing intergenerational dance classes running in communities (n = 11), Videos demonstrating intergenerational dance classes on websites (n = 2) | | McKinney Gibson, 2022 [74]; DU Dance NI, 2022 [75]; Dance Network Association, 2022 [76]; Intergenerational Dance: MIXER; Pereira-Stubbs; Identity Parade Green Candle Dance, 2014 [77]; Intergenerational Dance Project—Living Well, A Tale to Tell, 2022 [78]; Busuttil, 2021 [79]; Intergenerational Activity: (the Elders Dance), 2019 [80]; O'Connor, 2001 [81]; Magnetise: A Call for Home, 2023 [82]; Hoffman, 2019 [83]; Spurrier Dance Studio, 2019 [84]. |

studies that evaluated an intergenerational programme utilised a mixed methods design [57,60,64,67], two were quantitative [59,65], and five were qualitative [58,61–63,66].

## Characteristics of the literature

Tables 1 and 2 and S1 Table details the characteristics of the three groups of evidence sourced. The total sample size of the participants in the research studies was n = 973 (range n = 20, n = 572). Of the papers that reported sample size in the expert opinion/practice expertise, there was a sample size of n = 585 (range n = 8, n = 350). Sample size was not provided in the grey literature.

Of the papers that reported the gender of participants, [59–61,63,66], there were more female participants overall (a total of n = 67 female participants and n = 41 male participants).

The research studies were conducted in Singapore [61], the United Kingdom [57,62,63], the United States [58,67], Japan [59], Brazil [60], Canada [65,66], and Australia [64]. The expert opinion/practice expertise were largely from the United Kingdom [71,72,85], the United States [69,73].

**Table 2. Participant and dance genre characteristics in research papers n = 10 (excluding survey by D'Cunha et al. which had no dance or dose descriptors).**

| Authors | Sample size | Older population* | Younger population* | Dance Genre/ Intervention | Comparator/ Control | Outcome measures | Strength | Flexibility | Endurance | Balance | Frequency /week | Duration and Intensity | Dosage (frequency x duration) |
|---|---|---|---|---|---|---|---|---|---|---|---|---|---|
| Douse et al. 2020 [57] United Kingdom | 35 | 12 older adults (mean age 81–85) | 23 Teenage students (mean age 13–14) | Dance and photography | No intervention | The Basic Psychological Needs Satisfaction Scale, Positive and Negative Affect Scale, Social Well-Being Scale, Focus groups. | ✓ | ✓ | ✗ | ✓ | 1 | 90 mins x 11 weeks | 16.5 hours |
| Sherman, 1997 [58]. United States | 20 | 10 Profoundly deaf individuals aged 60+ living in a residence for older adults. | 10 9–12-year-old children enrolled in schools for the deaf | 'Dance' | No control | Interviews, field log. | ✗ | ✗ | ✗ | ✓ | 1 | Not reported | Not reported |
| Morita and Kobayashi, 2013 [59] Japan | 75 | 25 older adults aged 71–101. | 20 preschool children aged 5–6. | Performance-based IG** programme (singing and dancing) | Social-oriented programme | Intergenerational Exchanges Attitude Scale, "Smiling scale", Myers Research Institute Engagement Scale (MRI-ES), observation. | ✗ | ✗ | ✗ | ✗ | 1–2/month | 20–30 mins x 2 months | 2.67–8 hours (estimate) |
| Brandão et al. 2021 [60] Brazil | 48 | 28 older adults (10 people with stroke with expressive aphasia). Aged 60years+ | 20 undergraduate students | A series of online interventions including clowning, dancing, storytelling, and cooking | No control | Social Communication section of the Functional Assessment of Communication Skills for Adults (ASHA-FACS), Observation and evaluative conversations. | ✗ | ✗ | ✗ | ✓ | 1 | 60 mins x 12 weeks | 12 hours |
| Wu et al. 2023 [61] Singapore | 30 | 20 older adults from a senior activity centre | 10 student instructors (age 20–22). | Contemporary dance | No control | Semi-structured focus groups | ✓ | ✓ | ✗ | ✓ | 1 | 60 mins x 8 weeks | 8 hours |
| Farrer et al. 2022 [62] United Kingdom | 21 | 6 older adults from Generations Dancing | 8 teenage schoolchildren from Generations Dancing | Dance and photography programme | No control | Focus groups | ✓ | ✓ | ✗ | ✓ | 1 | 90 mins x 11 weeks | 16.5 hours |
| Jenkins et al. 2021 [63] United Kingdom | 65 | 9 Older adults living with a diagnosis of dementia or cognitive impairment involved in the Hear and Now project. | 9 younger people aged on average 11.5–13.4 involved in the Hear and Now project | Music and dance performance | No control | Focus groups | ✓ | ✓ | ✗ | ✓ | 4 weekends total | 4 sessions | Not reported |

*(Continued)*

**Table 2.** (Continued)

| Authors | Sample size | Older population* | Younger population* | Dance Genre/ Intervention | Comparator/ Control | Outcome measures | Strength | Flexibility | Endurance | Balance | Frequency /week | Duration and Intensity | Dosage (frequency x duration) |
|---|---|---|---|---|---|---|---|---|---|---|---|---|---|
| Rossberg Gempton and Poole, 2000 [65] Canada | 36 | 15 frail older adults aged 83 (mean) from intermediate care facilities | Children (n = 21; mean age:8) from rural schools | Creative dance | Children only class | N/A | ✓ | ✓ | ✓ | ✓ | Not reported | Not mentioned | Not reported |
| Rossberg Gempton et al. 1999 [66] Canada | 36 | 15 frail older adults aged 83 (mean) from intermediate care facilities | Children (n = 21; mean age:8) from rural schools | Creative dance | Children only class | Questionnaire for Participants in Dance/Movements Sessions, Staff Evaluation of Participant Response to Movement Sessions, video recording with coding | ✓ | ✓ | ✓ | ✓ | 2 | 30 mins x 12 weeks | 12 hours |
| Young | 35 | 12 grandparents who raise their grandchildren (aged 50+) | 23 grandchildren aged 5–17 | Zumba classes | No control | Short Portable Mental Status Questionnaire, Physical Activity Readiness Questionnaire, Timed Up and Go, Short Form-12 Health Survey, Centre for Epidemiologic Studies Depression Scale, The Rapid Assessment of Physical Activity, focus groups, interviews, observations. | ✓ | ✓ | ✓ | ✓ | 2 | 60 mins x 8 weeks | 16 hours |

✓: Yes ✗: No *No papers defined intergenerational. For the purpose of this review, intergenerational is two groups with at least one generational age gap. **IG = Intergenerational.

The age of the older adults in the research studies and expert opinion/practice expertise ranged from 55 to 101 years. All older cohorts in the research studies were described as "older adults". The older populations in the research evidence and grey literature were of similar cohorts, involving older dancers [74], older adults from rural nursing homes [81], long-term dementia care residents [83] and people living in the community [75,77–80,82,84,86,87].

Younger cohorts in the research studies primarily involved children in primary school, secondary school and in one instance, college (undergraduate) students. The grey literature evidence appeared to target younger age groups, involving more pre-school students [79], primary school students [78,80,81,83,87] and teenagers [75,77].

These programmes typically included an older adult cohort and young people who were unrelated. Young [67] included grandparents who were carers for their grandchildren, however, no other study included examined grandparents and grandchildren whose primary caregivers are not their grandparents.

Only two studies discussed dropout rates and adherence [57,60]. In Douse, Farrer [57], 33% of the older adults were reported as dropping out (n = 6, split evenly from control and experimental groups) and 21.7% of the younger cohort (n = 5, with four of these participants from the control group). Brandão, Bauer [60] had an adherence rate of 83.7%, with a total of ten dropouts. Six of these dropouts were involuntary due to reasons such as death (n = 3), no technical support available (n = 2), or worsening of delirium symptoms (n = 1).

## Definition of intergenerational

An aim of the scoping review was to identify the definition being used for 'intergenerational'. This was not defined by authors across all of the sources. Rather, specific age cohorts were picked, but there was no consistency in what was defined as intergenerational. Douse, Farrer [57] provided a definition for intergenerational practice, however this did not describe the actual generational relationships.

## Quality appraisal of research studies

The majority of the 11 studies appraised were shown to be of high quality on the MMAT with nine studies scoring 'Yes' on all sections [57,58,60–64,66,67]. See S3 File.

## Intervention content

Dance genres varied across the research studies. Two involved creative dance [65,66], two involved dance with photography [57,62], one was Zumba® based [67], one was contemporary dance [61], one was creating a dance for a performance [63] and three had multiple components (i.e., dance was just one aspect of the programme) [58–60]. The dance genre had greater variability in the available in the grey literature community classes, including imaginative and creative dance [86], online dance classes [78,82], dance creation [78,84], counterparty dance [81,83], ballet, and a combination of polka, waltz, schottische butterfly [80]. Two research papers were based on the same dance programmes, Generations Dancing [56,61], a creative dance programme [64,65]. The content was unclear in one paper [58] as the abstract and introduction describe a singing and dancing component, which is described further in methods or results/discussion, however, it was included as it did meet the original inclusion criteria.

Co-creation was described in four papers, where participants helped to create and choreograph their final performance [57,63,70,85]. Co-creation was included in two research studies, in the form of participants creating a performance [57,63]. In contrast, Brandão, Bauer [60] used a planned approach to the online sessions to ensure the activities were predictable and

familiar. The expert opinion/practice expertise papers spoke more about co-creation in dance performance and its benefits in exploring culture and community through dance [68,85]. The grey literature community classes focused on bringing the two generations together, rather than the specific dance elements.

Table 2 outlines the dance content and frequency, intensity, time and type (FITT) from each of the research studies insufficient information was provided for the other sources related to FITT. Many of the research studies focused on the process of choreography, creativity and the emotional nature of dance, therefore, the movements of the dances were not provided. Two of the authors on the review (a dance expert and physiotherapist) categorised the dance components based on the dance description provided in the research studies. The dance prescription for certain dance genres were not explicitly stated and therefore, the elements of the dances were up to interpretation. Strength, flexibility and balance were the most common components, with endurance being the least common component within the research studies. No interventional study compared the dance programmes to a conventional exercise programme. One study divided up the generations before bringing them together [57]. Other studies used the younger generation to teach or instruct the older generation [60,61]. The remainder of the studies included both generations together. Brandão, Bauer [60] based their intervention on multidimensional social interactions, in which dancing played one part. Wu, Yap [61] held their intervention in a recreational centre where other social and recreational programmes were held.

The duration of the intergenerational dance programme in the research studies ranged from four to 15-weeks, with 11–12 weeks being the most common duration. The frequency of the programmes ranged from twice weekly [66,67] to once/twice a month [59]. The duration of the sessions ranged from 30–90 minutes. Two studies did not state frequency [64,65]. Intensity was not stated in any of the studies. The variance in dosage ranged from 2.67–16.5 hours. Three of the research studies had a comparator intervention, two of these papers (based on the same programme) were children-only dance classes [65,66] and one was a social oriented programme [59], one study compared against no intervention [57]. The remainder of the research studies had no comparators [59,64–67].

## Outcomes

Overall, the outcomes used in the research studies were varied. Five research studies assessed their outcomes in both cohorts [57,60,62,63,65]. Young [67] assessed some outcomes in just older adults, and others in both cohorts. Four studies assessed outcomes in older adults only [58,59,61]. The main outcome assessed in the research studies was the experiences of participants, facilitators, or family or carers [57,58,61–63]. This was carried out through a variety of qualitative techniques including focus groups, interviews, evaluative conversations, observational notes and field logs. Three studies assessed social outcomes [57,59,66]. Douse, Farrer [57] used social outcomes including The Basic Psychological Needs Satisfaction Scale [88] and Positive and Negative Affect Scale [89]. Morita and Kobayashi [59] used the Intergenerational Exchanges Attitude Scale [90] and Rossberg-Gempton and Poole [65] looked at signs of cooperation/sense of belonging.

Quantitative outcomes included a range of physical outcomes such as body mass index, heart rate, and the Timed Up and Go [67].

One paper from the expert opinion/practise expertise literature described using logs, discussions and photographs to explore the experiences of participants [85]. The main aims in the community classes within the grey literature were to bring people together, exploring dance within these cohorts, and celebrating culture. The outcomes discussed in this context

related to the building relationships, shared connections, and the inclusivity of intergenerational dance, however these were not assessed objectively.

Quantitative findings were variable. Health benefits were reported quantitatively by Young [67]. Improvements, although not significant, were reported for health views (Short Form-12 Health Survey [91]), balance (timed up and go [92]), heart rate and depressive symptoms (Centre for Epidemiologic Studies Depression Scale [93]). Outcomes such as weight, body mass index and blood pressure did not show consistent improvements.

## Feasibility and implementation

Three studies assessed feasibility and acceptability of the programme using a mixed methods design involving qualitative interviews [57,60,67]. All three papers reported the programmes to be feasible and acceptable.

Location of the dance programme was found to be an important factor in the feasibility of these programmes [57,62,67,70,72] due to factors such as family members concern of participants' ability to mobilise to the location [53]. Benefits of specific locations were highlighted in two papers [62,67]. Farrer, Douse [62] found that a university location provided access to streams of funding and resources, while Young [67] remarked a community facility reduced the burden of organising staff and resources and allowed efforts to be focused on running the programme. However, inaccessibility and location led to dropouts due to the commute in one study [57]. Douse, Farrer [57] found that some family members worried about the older adults' ability to physically get to the location and three participants dropped out of the programme because of this. In line with the worry felt by family members in Douse, Farrer [57], Wu, Yap [61] recommended that future programmes should include risk factor identification and injury prevention strategies and older adult participants should be supported as beginner dancers. However, older adults are welcomed as dancers, with Prichard [68] noting that including dancers of different ages and generations can add to dance performances and older dancers generally bring experience and knowledge to dance.

## Qualitative findings

The qualitative findings from the qualitative (n = 5) and mixed methods (n = 4) papers were categorised into four main themes. These were *Attitudes towards the programmes*, *Relationship building*, *Self-expression and support through dance*, and *Health benefits*. Participants had largely positive attitudes towards the programmes and were able to build relationships with each other and facilitators. Dance was a medium to express emotion, while also helping participants to feel more mobile and healthier.

**Attitudes towards the programmes.** In five studies, many participants described how they felt positive emotions towards the programmes that they participated in [57,61,63,66,67]. Participants in two studies commented on how they looked forward to attending their programmes and were proud of their involvement [57,63]. Involvement in the community and relationship formation between groups was found to be a positive outcome in seven programmes [57,58,60,61,63,66,67].

**Relationship building.** Five papers found that relationships between the cohorts evolved throughout the programmes through the breakdown of stigma and stereotypes [57,58,60,61,63]. Moreover, the expert opinion/practice expertise papers noted that intergenerational dance programmes can break down barriers between generations and allow for a greater understanding of each other [73,85]. Intergenerational dance programmes were found to be beneficial for minority or marginalised groups due to the shared common ground between participants [58,70]. Notably, Sherman [58] remarked that bringing together two age

groups with similarities, in this case participants were from the deaf community, allowed them to find a common ground through the programme. The older adults in this programme developed a 'grandparenting' role towards the children.

**Self-expression and support through dance.** Dance was found to be a medium to express emotions in three studies [60,61,66]. This involved one participant using dance to express their grief [60], participants reminiscing and bonding through travel-based contemporary dance [61], and improvement of self-expression through music and imagery [66]. One author highlighted that intergenerational dance is an equalising medium for both generations, allowing for connections to be developed [85].

Participants from two studies commented on the loneliness experienced prior to their engagement in their programme [57,63]. Young [67], who investigated grandparent-grandchild dyads, found that that the grandparents felt more involved in their community during the programme.

**Health benefits.** Finally, five of the research papers qualitatively reported perceived health benefits the participants experienced [57,58,60,61,63]. These reported benefits comprised cognitive improvements [61], increased energy [58,60]. Jenkins, Farrer [63] found that participants, carers, and family members all noticed benefits in the participants following the programme. These benefits ranged from exercising vocal cords to increased range of motion. One participant commented how participation in the programme '[kept] us away from the doctors a lot, which is a good thing' [63].

## Discussion

This scoping review found limited empirical evidence for intergenerational dance programmes, specifically for cohorts with a generational gap. The scoping review identified a greater proportion of qualitative studies and studies evaluating psychosocial outcomes, with a dearth of papers examining the effects of intergenerational dance on physical health outcomes. This review has shown that whilst there are several intergenerational dance classes and sessions running in communities across the globe, there is insufficient information on the outcomes of these programmes as they are not being adequately documented or researched. Qualitatively, intergenerational dance was found to be a largely positive experience for participants, facilitators and family or carers, with the location of the programme being the main barrier to participation.

### Research design

The research evidence comprised two overall categories: interventional studies and expert opinion/practice expertise studies. Regarding methods, there was a greater proportion of qualitative studies that used a combination of interviews and focus groups. The qualitative papers in this review were generally high in quality and rigour. Qualitative research is important to understand the experiences of people (participants and facilitators) involved in these programmes, provide context and explain the rationale behind their behaviours [94,95]. Participants' experiences detailed through qualitative research methods can enable researchers to understand and interpret the context and behaviours that impact the success of intergenerational dance programmes [96]. In the context of intergenerational dance interventions, many outcomes cannot be adequately measured using quantitative measures, such as attitudes towards other age cohorts and how participants found participating in a new activity [97]. Qualitative research has been used in a dance programme for older adults previously, in addition to quantitative measures, which helped to enrich the dataset and highlight the benefits that did not necessarily reach statistical significance [98]. A qualitative evidence synthesis of

the studies in the area of intergenerational dance programmes would provide researchers designing future programmes insight into the needs and wants of participants to make an intervention meaningful to the target cohorts. Future research should use the qualitative research to guide the design and contents of their interventions.

To influence change at policy level, however, there is a need for quantitative research for richer data. Only two quantitative studies were found in this review, with a further four mixed-method papers. There were no randomised control trials or systematic reviews identified. Quantitative research is necessary to establish the effectiveness of these intergenerational dance programmes [99]. The quantitative elements in the included studies had greater variance in quality compared to the qualitative papers. Moreover, some of these studies, such as Morita and Kobayashi [59] utilised a quantitative methodology for outcomes that may have benefited from qualitative input. Several quantitative papers and systematic reviews have been carried out for dance in older adults [31,98,100] and young people [35,101,102] separately. The World Health Organisation found evidence to support the role of arts-based activities in health promotion and illness prevention across the lifespan [27], however, there are a lack of policies or guidelines specifically focusing on intergenerational cohorts and the benefits of these interactions. More quantitative studies are required in intergenerational dance so that systematic reviews can be conducted and eventually, guidelines or policies can be created around intergenerational dance.

**Intergenerational dance interventions**

The intergenerational research programmes largely targeted young people aged under 18 years and older adults aged 60+ years. This is beneficial as these age groups tend to report low levels of PA [11,103]. A previous study showed that during adolescence, there is a drop in PA levels, especially in girls, and there is an increase in sedentary time [104]. In older adults, a longitudinal study showed that older adults can spend up to 65% of their waking hours sedentary [105]. Both young people and older adults face challenges of loneliness and social isolation, with a U-shaped distribution of loneliness appearing across the lifespan [106,107]. However, there were a lack of studies in this review specifically including children aged eight years and younger. This age group may have issues with following instruction in a large group or have poor safety awareness, leading to an increased risk of injury [108]. Very young children and other vulnerable cohorts such as those from lower socioeconomic demographics have struggled to recover their PA levels following the COVID-19 pandemic [109]. However, children are typically most influenced by their parents' PA habits, therefore an intergenerational dance programme with older adults may not be as beneficial with this cohort [110]. In general, the groups targeted in the included intergenerational dance interventions were suitable ages and may benefit from such programmes to promote activity and combat social isolation.

The young people involved in the interventional studies were often facilitators, or were an aspect of the intervention, such as in Morita and Kobayashi [59]. Here, the children were performing for the older adults, and researchers observed the facial expressions of the older adults. This contrasted to the expert opinion/practice expertise where the experiences of the young participants were explored in more detail. This was possibly due to the aim of the studies and focus on older cohorts and their experiences in the interventional studies. Though some studies interviewed the young facilitators/instructors, others focused their assessments exclusively on the older participants. This excludes half of the possible dataset from the evaluation of the interventions. While young participants would possibly earn more physical benefits from a dance intervention [35], the social and emotional benefits from being an educator/facilitator/instructor should not be ignored. Assessing psychosocial outcomes such as mood and

wellbeing can give further insight into how young people are benefitting from such programmes. Young people can gain a greater understanding of older adults through instructing and it can improve the interactions between generations [111,112]. Considering the negative effects ageism has on society and the rise of intergenerational resentment [113,114], intergenerational dance programmes' ability to foster positive relationships between age cohorts is a promising step towards combating this prejudice. Future intergenerational dance programmes should ensure that both cohorts are assessed as participants within the interventions to evaluate outcomes and possible benefits for both groups.

The dance genre varied across the research articles and community classes (creative dance, ballet, contemporary dance). Some programmes included dance with other activities such as photography [57,59,60,62], but several studies did not specify the specific dance genre, or the movements involved in the activity [57,58,60,63]. The prescription of dance varied, and many of the research papers did not report the type of exercise involved in their dances i.e., strength, balance, or endurance (strength/balance/endurance). It is possible that dance was viewed as a social intervention in certain papers, neglecting the physical aspect. This may be beneficial for the participant as a previous qualitative paper noted that older adults often view exercise classes as a social outing, rather than purely for physical health [115]. However, it is important for the researchers to describe the dance and its movements as different dance genres can provide different benefits. For example, dance with sufficient and specific movements to challenge balance can elicit significant balance improvements for older adults [32]. Future studies should adequately report the exercise prescription in their dance programmes to highlight the dosage required to elicit reported improvements. Using a checklist such as the Template for Intervention Description and Replication (TIDieR) checklist [116] may help with describing the specifics of the interventions.

Co-creation was present in some of the research papers in the form of participants choreographing a performance. Co-creation, co-design, and co-production are popular participatory methods for creating public health and health promotion interventions [117]. Community-based participatory research and co-creation approaches have been used in intergenerational interventions [118–120]. The studies in this review that included co-creation gave participants opportunities to choreograph dances for a final performance, or they were given freedom to explore certain movements within the sessions themselves. Many of these participants felt they could express themselves through their movements. Co-creation is recommended in community settings [121] and in dance, it allows participants to explore both the creation of the dance and the physical surroundings [122,123]. Within health research, co-creation can have several benefits including empowering participants and relationship building [124]. It does this through giving a voice to those who the interventions are designed for, identifying their needs, and building a trusting relationship between researchers and the public [117]. While co-creation is a feasible method for creating an intervention, there are no guidelines or frameworks to guide the process for researchers [125]. Failing to consider participants' opinions through co-creation can lead to higher attrition rates [121]. Based on the findings in this review, choreography appears to be a good way of implementing co-creation into intergenerational dance programmes however, additional co-creation and co-design methods should be considered in future interventions. This may include choosing the location and timing of the intervention, the music playlist for the dances, and other non-dance related activities in the programme.

## Social interaction

Loneliness was a motivating factor for some of the older adults to participate in the intergenerational dance programmes [57,63]. This was possibly due to the older adults wanting to meet

likeminded people within their community. PA can play a role in reducing loneliness, however this is dependent on the relationships with others during the activity [126]. The social nature of group exercise classes has been shown to help mitigate loneliness in certain older adult groups [19]. It should be noted that feeling lonely can reduce motivation to participate in PA [127]. Therefore, it is important to consider the enjoyment and accessibility of the programmes to ensure continued attendance and encourage people to join such interventions. Dance interventions can boost social engagement in older adults [128] and intergenerational interventions can reduce social isolation and loneliness [129,130]. This review highlighted the importance intergenerational relationships formed through dance have for both older and younger generations.

A sense of community and relationship building was mentioned in the qualitative aspects of several papers [63,64,67]. This mirrored the expert opinion/practice expertise papers that highlighted the bonds that can be formed through intergenerational dance. Older adults typically do not have many opportunities to socialise with each other, therefore, the intergenerational dance programmes provided a unique experience for both. A systematic review has shown that intergenerational interactions within interventions can build communities and improve health-related outcomes in older adults [131]. The intervention run by Brandão, Bauer [60] occurred during the initial Covid-19 lockdown and participants enjoyed being able to see each other and create a virtual community. While this contrasts research that found online interactions to be less effective at improving loneliness [132], it may be an option for those living in more rural areas where transport may be a barrier. In-person settings are beneficial to involve participants in their community. Intergenerational dance programmes may be an opportunity for communities to reduce social isolation experienced by the young and old.

## Health outcomes

Young [67] was the only study to assess a battery of physical outcome measures, finding benefits in mobility and health-related quality of life in older adults. While not objectively assessed, improvements in self-reported health outcomes were mentioned by some participants within the qualitative studies, such as improved energy levels, cognition, and mobility. The papers in this review assessed outcomes based on their aims, which were mainly psychosocial, however dance is a safe and acceptable form of exercise; therefore, physical improvements would be expected depending on the dance dosage had they been assessed. Previous research indicated that dance can be equally effective, if not more, than conventional PA interventions for older adults [32] and can provide physical and psychological benefits for young people [35], hence studies should look at the physical effects of intergenerational dance. More research studies assessing the effectiveness of intergenerational dance on physical outcomes are needed so that future systematic reviews can be conducted, similar to existing syntheses for older adults and young people separately [32,36]. This can hopefully help to inform policies and frameworks in the future.

To guide the outcome measures assessed in future studies, the qualitative findings can be used as a guide what is meaningful and important for the included participants [133]. The possibility of developing a core outcome set should be considered for future intergenerational dance studies to maximise comparability in future research [134].

## Barriers and facilitators

This scoping review identified important pragmatic considerations for future studies. A barrier to participation is the location of the intervention. The inaccessibility of the location used by Douse, Farrer [57] resulted in three participants dropping out. When including frail older

adults in such an intervention, the venue should be accessible to those with mobility aids and disability friendly [135]. Accessible locations and convenient transport are facilitators for adults to attend fitness centres [136]. Previous studies have highlighted that the location of the interventions is a major facilitator and community centres are accessible and convenient for participants [137,138]. To ensure the feasibility of intergenerational dance programmes, the needs and abilities of participants must be considered. Locations of the interventions should be accessible to those of all abilities. Moreover, the overall quality of intergenerational programmes is important for their success. Future programmes should be well designed using the recommendations from previous intergenerational programmes.

The interventionalists ideally should have a good relationship with the host organisation, as well as the participants [139]. Inclusion, social and socio-economic circumstances were highlighted by Young [67], where participation was affected by circumstances outside of the participants' control, such as grandparents' custody of the grandchildren. A diet and PA intergenerational study for vulnerable Appalachian residents failed to elicit significant behaviour change due to participants' heavy health burdens and resource scarcity [119]. Despite the evidence-based approach, the intervention was not as successful as expected due to pre-existing barriers. Therefore, it is important to consider multiple behaviour change techniques and levels of influence to conduct a successful intervention with high adherence for people from certain disadvantaged groups. To recruit people from lower socioeconomic statuses, active in-person recruitment strategies, social media, and liaising with the social network comprising of caregivers and experts should be utilised, however challenges remain [140].

## Strengths and limitations

To the best of our knowledge, this is the first review of its kind to look at intergenerational dance programmes, specifically with a generational gap. A systematic approach was taken towards all stages and the JBI methodology for scoping reviews was followed to ensure its replicability. An expert librarian was contacted to assist with the search strategies. Two independent reviewers who were blinded carried out the paper selection process.

Despite the robust search across several interdisciplinary databases, there were a lack of suitable studies. As the review question was broad, there were many different types of sources included. This made it difficult to appraise the quality of the papers and only the interventional studies were appraised using the MMAT. However, quality appraisal is not a requirement in a scoping review. This was conducted by the reviewers to give an indication of the quality of available studies.

## Conclusion

The results of this scoping review highlight the lack of empirical evidence to support intergenerational dance programmes. The included papers had considerable variance in methods, intervention, quality and outcomes. There is a bias towards qualitative research in this area. The qualitative papers highlighted the positive feelings people associate with such programmes and the self-reported benefits they have experienced through participation. The expert opinion/practice expertise research papers discussed theories and methods to choreographing intergenerational dance, and the benefits of expressing oneself through this medium. Based on the gaps identified in this review, there is a need for more high-quality research, including primary research with a quantitative focus. The outcome measures assessed in studies should be relevant and meaningful to participants, combining physical outcomes and psychosocial outcomes. Additionally, this review has highlighted that while there are many classes running in communities, these are not being formally documented in many cases. Researchers should

consider the use of a homogenous definition for "intergenerational" when designing studies. There is a need for future research to focus on larger-scale quantitative or mixed methods to examine the effectiveness of intergenerational dance studies.

## Supporting information

**S1 File. PRISMA-ScR checklist.**
(PDF)

**S2 File. Example search strategy.**
(DOCX)

**S3 File. Appraisal tables.**
(DOCX)

**S1 Table. Data Extraction table.**
(XLSX)

## Acknowledgments

We would like to thank the authors of the original research included in this review.

## Author Contributions

**Conceptualization:** Siobhán O'Reilly, Orfhlaith Ní Bhriain, Amanda M. Clifford.

**Funding acquisition:** Siobhán O'Reilly.

**Supervision:** Orfhlaith Ní Bhriain, Amanda M. Clifford.

**Validation:** Siobhán O'Reilly, Sarah Dillon.

**Visualization:** Siobhán O'Reilly.

**Writing – original draft:** Siobhán O'Reilly.

**Writing – review & editing:** Siobhán O'Reilly, Orfhlaith Ní Bhriain, Sarah Dillon, Amanda M. Clifford.

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
