## [Decision Letter · Decision Letter 0]

24 Jun 2024

PONE-D-24-12051A comprehensive scoping review of intergenerational dance programmes for cohorts with a generational gap.PLOS ONE

Dear Dr. O'Reilly,

Thank you for submitting your manuscript to PLOS ONE. After careful consideration, we feel that it has merit but does not fully meet PLOS ONE’s publication criteria as it currently stands. Therefore, we invite you to submit a revised version of the manuscript that addresses the points raised during the review process.

We look forward to receiving your revised manuscript.

Kind regards,

Jindong Chang, Ph.D.

Academic Editor

PLOS ONE

Journal Requirements:

Additional Editor Comments:

Dear Dr. O'Reilly,

Thank you for choosing our journal to submit your manuscript. After the peer review process, it has been determined that there are a few minor issues that require your attention for revision. The reviewers' feedback is as follows:

######

reviewer 1：An important and interesting topic explored with solid methodology. Well done to the authors for undertaking this study. Only a few minor edits to help with clarity and a couple of spots where I hope there can be more depth of discussion to strengthen this paper further.

Line 74 missing word? "...as a positive and enjoyable experience is a motivating for older adults and youths alike..."

Methods

Line 165 missing citation for Endnote software

Results

Please clarify the total number of papers. Line 194 states "19 research evidence papers: eleven interventional studies, seven expert opinion..." This adds up to eighteen. Then in Table 1, the row for Mixed methods studies states an N=4, specifically the row for pretest-posttest trials n=3 but only 2 studies are cited. So only 10 interventional studies total.

Line 265 typo "been" either remove or replace with 'being'

Table 2 define IG

The dropout rate mentioned in Line 322/323 is hidden at the moment within the feasibility paragraph discussing the impact of location. It would be better to make this more prominent in the sample size reporting earlier in your results and also get the breakdown of which studies reported attrition from the study versus lack of attendance in the dance program.

Line 332 Minor typo capitalise the word "self-expression" for consistency with your other themes.

Line 349 minor typo replace the word "do" with "due" in the line "...for minority or marginalised groups do to the shared common ground..."

Discussion

Line 407 please reword for clarity "this age group". do you mean the young people under 18 years or the older adults aged 60+ years, or both?

Lines 413-414 the reference to younger children

Paragraph lines 423-431 I'd like to have seen more elaboration about the outcomes and proposed potential benefits for the young people in the interventional studies that you would like to see, and if their role as facilitators and/or participants would impact the outcomes.

Line 434 typo "with" replace with "but"

Lines 435, 436-437, and 438 you have repeated the same phrase "the specific movements involved in the dances were not always described" across three consecutive sentences. I am not sure if you mean different concepts here and need to clarify your wording, or if you need to remove the repetition.

Line 448 excellent idea providing a tool for future researchers to use!

Paragraph lines 450-462 co-creation could potentially involve more than just the choreography aspect. I think this idea could be explored in a deeper way and perhaps link to co-design and co-creation for health promotion activities and the potential for adherence and success. Your concepts around community success and the benefits of empowering participants and relationship building should be elaborated upon. What other ways can co-creation be implemented?

Conclusion

Line 558 typo "combing" replace with "combining.

########

reviewer 2：Thanks for the opportunity to review this interesting and well written paper. The results of this scoping review highlight the lack of empirical evidence to support intergenerational dance programmes. The included papers had considerable variance in methods, intervention, quality, and outcomes. There is a bias towards qualitative research in this area. The qualitative papers highlighted the positive feelings people associate with such programmes and the self-reported benefits they have experienced through participation. The expert opinion/practice expertise research papers discussed theories and methods of choreographing intergenerational dance and the benefits of expressing oneself. I have a few questions and suggestions for consideration:

1. What was the study question(s)? You mentioned "The aim of this scoping review is to identify and map the literature on dance programmes for intergenerational cohorts."

2. Is it possible to list the outcome measures in reviewed article in table 2.

3. Both qualitative and quantitative research contribute rigorously to the evidence base in different but equally valuable ways. Can you include a discussion on how qualitative findings can provide deep insights into personal experiences and the social context of intergenerational dance programmes, which can enrich the understanding of the impact of intergenerational dance on older people and young people. Provide more detailed examples and cite the paper you reviewed.

Please have the authors address the following specific points:

Correct minor typographical errors and missing words.

Clarify the total number of papers and the categorization of study types.

Expand the discussion on the dropout rate, the role of young people, and the concept of co-creation.

Ensure consistency in thematic presentation and avoid repetition of phrases.

Clearly state the study question(s) and include a discussion on the contribution of qualitative and quantitative research to the evidence base.

I look forward to seeing the revised manuscript and believe that with these improvements, it will make a valuable contribution to the field.

Sincerely,

Jindong Chang

Academic Editor

Reviewers' comments:

Reviewer's Responses to Questions

**Comments to the Author**

1. Is the manuscript technically sound, and do the data support the conclusions?

Reviewer #1: Yes

Reviewer #2: Yes

2. Has the statistical analysis been performed appropriately and rigorously? 

Reviewer #1: N/A

Reviewer #2: N/A

3. Have the authors made all data underlying the findings in their manuscript fully available?

Reviewer #1: Yes

Reviewer #2: Yes

4. Is the manuscript presented in an intelligible fashion and written in standard English?

Reviewer #1: Yes

Reviewer #2: Yes

5. Review Comments to the Author

Reviewer #1: An important and interesting topic explored with solid methodology. Well done to the authors for undertaking this study. Only a few minor edits to help with clarity and a couple of spots where I hope there can be more depth of discussion to strengthen this paper further.

Line 74 missing word? "...as a positive and enjoyable experience is a motivating for older adults and youths alike..."

Methods

Line 165 missing citation for Endnote software

Results

Please clarify the total number of papers. Line 194 states "19 research evidence papers: eleven interventional studies, seven expert opinion..." This adds up to eighteen. Then in Table 1, the row for Mixed methods studies states an N=4, specifically the row for pretest-posttest trials n=3 but only 2 studies are cited. So only 10 interventional studies total.

Line 265 typo "been" either remove or replace with 'being'

Table 2 define IG

The dropout rate mentioned in Line 322/323 is hidden at the moment within the feasibility paragraph discussing the impact of location. It would be better to make this more prominent in the sample size reporting earlier in your results and also get the breakdown of which studies reported attrition from the study versus lack of attendance in the dance program.

Line 332 Minor typo capitalise the word "self-expression" for consistency with your other themes.

Line 349 minor typo replace the word "do" with "due" in the line "...for minority or marginalised groups do to the shared common ground..."

Discussion

Line 407 please reword for clarity "this age group". do you mean the young people under 18 years or the older adults aged 60+ years, or both?

Lines 413-414 the reference to younger children

Paragraph lines 423-431 I'd like to have seen more elaboration about the outcomes and proposed potential benefits for the young people in the interventional studies that you would like to see, and if their role as facilitators and/or participants would impact the outcomes.

Line 434 typo "with" replace with "but"

Lines 435, 436-437, and 438 you have repeated the same phrase "the specific movements involved in the dances were not always described" across three consecutive sentences. I am not sure if you mean different concepts here and need to clarify your wording, or if you need to remove the repetition.

Line 448 excellent idea providing a tool for future researchers to use!

Paragraph lines 450-462 co-creation could potentially involve more than just the choreography aspect. I think this idea could be explored in a deeper way and perhaps link to co-design and co-creation for health promotion activities and the potential for adherence and success. Your concepts around community success and the benefits of empowering participants and relationship building should be elaborated upon. What other ways can co-creation be implemented?

Conclusion

Line 558 typo "combing" replace with "combining"

Reviewer #2: Thanks for the opportunity to review this interesting and well written paper. The results of this scoping review highlight the lack of empirical evidence to support intergenerational dance programmes. The included papers had considerable variance in methods, intervention, quality, and outcomes. There is a bias towards qualitative research in this area. The qualitative papers highlighted the positive feelings people associate with such programmes and the self-reported benefits they have experienced through participation. The expert opinion/practice expertise research papers discussed theories and methods of choreographing intergenerational dance and the benefits of expressing oneself. I have a few questions and suggestions for consideration:

1. What was the study question(s)? You mentioned "The aim of this scoping review is to identify and map the literature on dance programmes for intergenerational cohorts."

2. Is it possible to list the outcome measures in reviewed article in table 2.

3. Both qualitative and quantitative research contribute rigorously to the evidence base in different but equally valuable ways. Can you include a discussion on how qualitative findings can provide deep insights into personal experiences and the social context of intergenerational dance programmes, which can enrich the understanding of the impact of intergenerational dance on older people and young people. Provide more detailed examples and cite the paper you reviewed.

6. PLOS authors have the option to publish the peer review history of their article (what does this mean?). If published, this will include your full peer review and any attached files.

Reviewer #1: No

Reviewer #2: **Yes: **Lillian Hung

---

## [Author Response · Author response to Decision Letter 0]

14 Aug 2024

Thank you to the reviewers for your helpful and insightful feedback. The comments have been addressed and the responses are outlined below:

Line 74 missing word? "...as a positive and enjoyable experience is a motivating for older adults and youths alike...": This has been changed to “is a motivator”. (Manuscript line 74)

Line 165 missing citation for Endnote software: Citation has been included. (Manuscript line 168)

Please clarify the total number of papers. Line 194 states "19 research evidence papers: eleven interventional studies, seven expert opinion..." This adds up to eighteen. Then in Table 1, the row for Mixed methods studies states an N=4, specifically the row for pretest-posttest trials n=3 but only 2 studies are cited. So only 10 interventional studies total: This has been amended. There were 18 research evidence papers: 2 quantitative, 5 qualitative, 4 mixed methods (The citation for Young et al. (2014) has been included, and 7 expert opinion. (Manuscript line 196)

Line 265 typo "been" either remove or replace with 'being': Replaced with “being”. (Manuscript line 274)

Table 2 define IG: No definitions were reported in the included papers for intergenerational. A note has been added to the table with the authors’ description for the purpose of this review. (Table 2)

The dropout rate mentioned in Line 322/323 is hidden at the moment within the feasibility paragraph discussing the impact of location. It would be better to make this more prominent in the sample size reporting earlier in your results and also get the breakdown of which studies reported attrition from the study versus lack of attendance in the dance program: Additional information (paragraph lines 229-234) has been included to discuss dropouts/adherence. 

“Only two studies discussed dropout rates and adherence (57, 60). In Douse, Farrer (57), 33% of the older adults were reported as dropping out (n=6, split evenly from control and experimental groups) and 21.7% of the younger cohort (n=5, with four of these participants from the control group). Brandão, Bauer (60) had an adherence rate of 83.7%, with a total of ten dropouts. Six of these dropouts were involuntary due to reasons such as death (n=3), no technical support available (n=2), or worsening of delirium symptoms (n=1).”

Line 331/332 has been removed to avoid repetition 

Line 332 Minor typo capitalise the word "self-expression" for consistency with your other themes: This typo has been amended. (Manuscript line 341)

Line 349 minor typo replace the word "do" with "due" in the line "...for minority or marginalised groups do to the shared common ground...": This typo has been amended. (Manuscript line 358) 

Line 407 please reword for clarity "this age group". do you mean the young people under 18 years or the older adults aged 60+ years, or both?: This was in reference to both age groups and has been reworded to “This is beneficial as these age groups tend to report low levels of PA (11, 102).” (Manuscript line 428)

Lines 413-414 the reference to younger children: “Younger” has been removed. (Manuscript line 435)

Paragraph lines 423-431 I'd like to have seen more elaboration about the outcomes and proposed potential benefits for the young people in the interventional studies that you would like to see, and if their role as facilitators and/or participants would impact the outcomes: To address the possible benefits of young people, lines 450 to 462 have been added.

“Though some studies interviewed the young facilitators/instructors, others focused their assessments exclusively on the older participants. This excludes half of the possible dataset from the evaluation of the interventions. While young participants would possibly earn more physical benefits from a dance intervention (34), the social and emotional benefits from being an educator/facilitator/instructor should not be ignored. Assessing psychosocial outcomes such as mood and wellbeing can give further insight into how young people are benefitting from such programmes. Young people can gain a greater understanding of older adults through instructing and it can improve the interactions between generations (110, 111). Considering the negative effects ageism has on society and the rise of intergenerational resentment (112, 113), intergenerational dance programmes’ ability to foster positive relationships between age cohorts is a promising step towards combating this prejudice. Future intergenerational dance programmes should ensure that both cohorts are assessed as participants within the interventions to evaluate outcomes and possible benefits for both groups.”

Line 434 typo "with" replace with "but": This has been amended. (Manuscript line 464)

Lines 435, 436-437, and 438 you have repeated the same phrase "the specific movements involved in the dances were not always described" across three consecutive sentences. I am not sure if you mean different concepts here and need to clarify your wording, or if you need to remove the repetition: Line 465 has been removed (“Consistent descriptions were not used…”) and the subsequent sentence has been expanded to clarify that the papers did not report exercise types (strength/balance/endurance) in addition to not describing the actual dance movements. (Manuscript line 469)

“several studies did not specify the specific dance genre, or the movements involved in the activity (57, 58, 60, 63). The prescription of dance varied, and many of the research papers did not report the type of exercise involved in their dances i.e., strength, balance, or endurance.”

Line 448 excellent idea providing a tool for future researchers to use!: Thank you.

Paragraph lines 450-462 co-creation could potentially involve more than just the choreography aspect. I think this idea could be explored in a deeper way and perhaps link to co-design and co-creation for health promotion activities and the potential for adherence and success. Your concepts around community success and the benefits of empowering participants and relationship building should be elaborated upon. What other ways can co-creation be implemented?: Thank you for this comment. There are many benefits associated with co-creating and co-designing an intervention. While choreography was most commonly seen in the included papers, there are many alternate ways to cocreate an intervention and it would be good to see the co-creation of dance interventions expanded to other aspects of the programme such as location and timing of the programme, the contents of the sessions, and the music. This has been discussed in paragraph lines 481-498.

“Community-based participatory research and co-creation approaches have been used in intergenerational interventions (117-119). The studies in this review that included co-creation gave participants opportunities to choreograph dances for a final performance, or they were given freedom to explore certain movements within the sessions themselves. Many of these participants felt they could express themselves through their movements. Co-creation is recommended in community settings (120) and in dance, it allows participants to explore both the creation of the dance and the physical surroundings (121, 122). Within health research, co-creation can have several benefits including empowering participants and relationship building (123). It does this through giving a voice to those who the interventions are designed for, identifying their needs, and building a trusting relationship between researchers and the public (116). While co-creation is a feasible method for creating an intervention, there are no guidelines or frameworks to guide the process for researchers (124). Failing to consider participants’ opinions through co-creation can lead to higher attrition rates (120). Based on the findings in this review, choreography appears to be a good way of implementing co-creation into intergenerational dance programmes however, additional co-creation and co-design methods should be considered in future interventions. This may include choosing the location and timing of the intervention, the music playlist for the dances, and other non-dance related activities in the programme.”

Line 558 typo "combing" replace with "combining: This typo has been amended (Manuscript line 596).

What was the study question(s)? You mentioned "The aim of this scoping review is to identify and map the literature on dance programmes for intergenerational cohorts.": The research question is what is the evidence for intergenerational dance programmes with an intergenerational gap between participants? This has been clarified in lines 107-108

Is it possible to list the outcome measures in reviewed article in table: A column has been added to include outcome measures in Table 2.

Both qualitative and quantitative research contribute rigorously to the evidence base in different but equally valuable ways. Can you include a discussion on how qualitative findings can provide deep insights into personal experiences and the social context of intergenerational dance programmes, which can enrich the understanding of the impact of intergenerational dance on older people and young people. Provide more detailed examples and cite the paper you reviewed: In line with this comment, we have now added the following section to the Discussion Section, Lines 399 to 410 

“Participants’ experiences detailed through qualitative research methods can enable researchers to understand and interpret the context and behaviours that impact the success of intergenerational dance programmes (95). In the context of intergenerational dance interventions, many outcomes cannot be adequately measured using quantitative measures, such as attitudes towards other age cohorts and how participants found participating in a new activity (96). Qualitative research has been used in a dance programme for older adults previously, in addition to quantitative measures, which helped to enrich the dataset and highlight the benefits that did not necessarily reach statistical significance (97). A qualitative evidence synthesis of the studies in the area of intergenerational dance programmes would provide researchers designing future programmes insight into the needs and wants of participants to make an intervention meaningful to the target cohorts. Future research should use the qualitative research to guide the design and contents of their interventions.”

---

## [Decision Letter · Decision Letter 1]

18 Sep 2024

PONE-D-24-12051R1A comprehensive scoping review of intergenerational dance programmes for cohorts with a generational gap.PLOS ONE

Dear Dr. O'Reilly, Thank you for submitting your manuscript to PLOS ONE. After careful consideration, we feel that it has merit but does not fully meet PLOS ONE’s publication criteria as it currently stands. Therefore, we invite you to submit a revised version of the manuscript that addresses the points raised during the review process.

Please have the author carefully consider the issues raised by the reviewer and address the reviewer's concerns.

We look forward to receiving your revised manuscript.

Kind regards,

Jindong Chang, Ph.D.

Academic Editor

PLOS ONE

Journal Requirements:

Reviewers' comments:

Reviewer's Responses to Questions

**Comments to the Author**

1. If the authors have adequately addressed your comments raised in a previous round of review and you feel that this manuscript is now acceptable for publication, you may indicate that here to bypass the “Comments to the Author” section, enter your conflict of interest statement in the “Confidential to Editor” section, and submit your "Accept" recommendation.

Reviewer #1: (No Response)

2. Is the manuscript technically sound, and do the data support the conclusions?

Reviewer #1: Yes

3. Has the statistical analysis been performed appropriately and rigorously? 

Reviewer #1: Yes

4. Have the authors made all data underlying the findings in their manuscript fully available?

Reviewer #1: Yes

5. Is the manuscript presented in an intelligible fashion and written in standard English?

Reviewer #1: Yes

6. Review Comments to the Author

Reviewer #1: Thank you to the authors for addressing all of the comments thoroughly and thoughtfully.

Only one edit was not addressed completely: Table 2 the definition of IG. In row 3 Morita and Kobayashi, 2012, the description of the program states IG. After the abbreviation IG please add '*', then the definition below the table should read "IG = intergenerational". The rest of the explanation may/may not need to be included if you wish as this is described in the text.

Great job on this lovely paper.

7. PLOS authors have the option to publish the peer review history of their article (what does this mean?). If published, this will include your full peer review and any attached files.

Reviewer #1: No

---

## [Author Response · Author response to Decision Letter 1]

19 Sep 2024

Dear Reviewer,

Thank you very much for your kind feedback.

Table 2 row three has been amended to include “**”, with "IG = intergenerational" included below the table. 

Kind regards,

Siobhán

---

## [Editor Report · Decision Letter 2]

23 Sep 2024

A comprehensive scoping review of intergenerational dance programmes for cohorts with a generational gap.

PONE-D-24-12051R2

Dear Dr. Siobhán O’Reilly,

We’re pleased to inform you that your manuscript has been judged scientifically suitable for publication and will be formally accepted for publication once it meets all outstanding technical requirements.

Kind regards,

Jindong Chang, Ph.D.

Academic Editor

PLOS ONE

---

## [Editor Report · Acceptance letter]

17 Oct 2024

PONE-D-24-12051R2 

PLOS ONE

Dear Dr. O'Reilly, 

I'm pleased to inform you that your manuscript has been deemed suitable for publication in PLOS ONE. Congratulations! Your manuscript is now being handed over to our production team.

Kind regards, 

on behalf of

Dr. Jindong Chang 

Academic Editor

PLOS ONE